# Exogenous Melatonin Alters Stomatal Regulation in Tomato Seedlings Subjected to Combined Heat and Drought Stress through Mechanisms Distinct from ABA Signaling

**DOI:** 10.3390/plants12051156

**Published:** 2023-03-03

**Authors:** Nikolaj Bjerring Jensen, Carl-Otto Ottosen, Rong Zhou

**Affiliations:** 1Department of Food Science, Plant, Food & Climate, Aarhus University, Agro Food Park 48, DK-8200 Aarhus N, Denmark; 2College of Horticulture, Nanjing Agricultural University, Nanjing 210095, China

**Keywords:** stomata, heat, drought, combined stress, ABA, melatonin, ROS, g_s_

## Abstract

The understanding of stomatal regulation in climate stress is essential for ensuring resilient crops. The investigation of the stomatal regulation in combined heat and drought stress aimed to link effects of exogenous melatonin on stomatal conductance (g_s_) and its mechanistic interactions with ABA or ROS signaling. Melatonin-treated and non-treated tomato seedlings were subjected to moderate and severe levels of heat (38°C for one or three days) and drought stress (soil relative water content of 50% or 20%) applied individually and in combination. We measured g_s_, stomatal anatomy, ABA metabolites and enzymatic ROS scavengers. The stomata in combined stress responded predominantly to heat at soil relative water content (SRWC) = 50% and to drought stress at SRWC = 20%. Drought stress increased ABA levels at severe stress, whereas heat stress caused an accumulation of the conjugated form, ABA glucose ester, at both moderate and severe stress. The melatonin treatment affected g_s_ and the activity of ROS scavenging enzymes but had no effect on ABA levels. The ABA metabolism and conjugation of ABA might play a role in stomatal opening toward high temperatures. We provide evidence that melatonin increases g_s_ in combined heat and drought stress, but the effect is not mediated through ABA signaling.

## 1. Introduction

Global warming due to human activities is projected to reach 1.5 °C between 2030 and 2052. This will impact on the climate systems, causing changes in precipitation patterns, an increased frequency of heat waves and a risk of regional water scarcity [1]. Heat and drought stress have negative effects on the crop performance and yield, and these abiotic stress factors are considered as some of the main challenges in the future of agriculture [2]. 

While it has been the golden standard for decades in plant physiology research to study responses to abiotic stresses applied individually, it has become clear that the combination of different abiotic stresses does not only have interactions, but also causes unique responses when combined that cannot be extrapolated from the responses to the individual stress factors [3]. This insight has opened up the relatively new research field of combining abiotic stress factors, and recent studies have demonstrated effects of combined stress responses distinct from the individual stress responses at both physiological, transcriptional and metabolic levels [4,5,6].

The combination of heat and drought stress has several implications on plant physiology in both the acute stress phase and the long-term adaptation to stress [7]. In the acute stress phase, both heat and drought stress can trigger changes in the metabolism of reactive oxygen species (ROS). During drought stress, ROS is mainly generated in the chloroplasts by increased rates of photorespiration and the Mehler reaction as a result of an imbalance between electron release and acceptance caused by reduced carbon fixation due to a decrease in the stomatal aperture [8]. The increase in ROS at high temperature is caused by the uncoupling or disruption of membrane-bound electron transporters in the mitochondria, causing a reduction of O_2_ to ROS [9] and impairment of the electron flow in the light reaction of photosynthesis in chloroplasts due to heat-induced damage to photosystem II [8]. To counterbalance the increased ROS production and oxidative stress, plants can increase the levels of non-enzymatic antioxidants and the expression of ROS scavenging enzymes such as superoxide dismutase (SOD) and catalase (CAT) that scavenge the superoxide radical and hydrogen peroxide, respectively; along with ascorbate peroxidase (APX) and glutathione reductase (GR) of the ascorbate–glutathione cycle [10]. Apart from having direct damaging effects to cell components, ROS also play signaling roles during stress by promoting the expression of stress-related genes and integrating with hormonal signaling [11,12]. 

Stomata are pores in the plant epidermis that are bordered by two guard cells with the ability to control the pore opening to regulate the gas exchange of CO_2_ and water vapor across the leaf surface. Stomatal properties such as the stomata size, pore area and stomatal density affects the stomatal conductance (g_s_) of the leaf [13]. When the g_s_ is high, transpiration can occur at high rates to decrease the leaf temperature through evaporative cooling [14]. One of the aspects of leaf stress physiology where heat and drought gives opposing cues is in stomatal regulation. Drought will signal the stomata to close in order to reduce transpiration and conserve water, whereas heat will promote stomatal opening to allow for an increased leaf cooling [15]. The hormone abscisic acid (ABA) is considered to be the key regulator in stomatal closing at drought [16], and the signaling mechanisms by which it acts to regulate the stomatal aperture have been uncovered in great detail [17]. The foliar level of ABA is determined by both the transport of root-synthesised ABA to the leaf acting as a root-to-shoot signal to close stomata and the rate of ABA synthesis and breakdown in the leaf, as well as the conjugation with glucose to form the inactive storage from, ABA glucose ester (ABA-GE) [18,19,20].

Less is known about the signaling mechanisms behind the response of stomatal opening to high temperatures. However, two distinct pathways have recently been suggested: a phototropin-dependent pathway and a phototropin-independent pathway that involves ROS as a signaling component [21]. Drought-stress-induced stomatal closing has a direct influence on parameters such as a reduced cellular water activity, reduced intercellular CO_2_ concentration (c_i_) and increased leaf temperature, which, in turn, can trigger downstream negative effects on photosynthesis, change the metabolic homeostasis and increase cellular oxidative stress. Such effects might be more pronounced in combination with heat stress [9,22,23]. Therefore, the understanding of stomatal regulation in combined heat and drought stress is central for deducing the mechanisms of plant tolerance to environments where these stress factors co-occur.

Melatonin has been shown to be present in smaller quantities in a wide range of plants, including common crop plants [24]. In tomato, melatonin has been reported to be naturally present at levels of 1.5–140 ng/g FW dependent on environmental conditions [25,26]. The molecular targets of melatonin have not been fully uncovered. Recently, a membrane-localized NADPH oxidase, respiratory burst oxidase homologue (RBOH) and a plasma membrane-localized protein with receptor-like topology (CAND2/PMTR1) have been identified as possible melatonin receptors in plants [27,28]. The detailed molecular mechanisms and signal transduction pathways induced by melatonin in the context of various stress situations, however, remains largely undescribed. 

Many studies reporting biochemical and physiological effects in response to the exogenous application of melatonin suggest that melatonin does play various roles, with the potential to mitigate negative effects of abiotic stresses. It seems well established that exogenous melatonin can act to reduce ROS through the upregulation of radical scavenging enzymes [29,30,31,32]. The exogenous application of melatonin has also been shown to increase g_s_ in plants exposed to various kinds of stress, such as in wheat recovering from cold stress [33] and in drought-stressed apple [34]. In recent reviews, it has been proposed that melatonin plays regulating roles in phytohormonal systems, such as ABA regulation [29,35]. It can be speculated that melatonin could exert effects on ABA metabolism and stomatal regulation through the modulation of ROS signaling, since the accumulation of ABA is induced by the generation of ROS and oxidative stress [36,37].

However, no studies have investigated the effects of exogenous melatonin on stomatal regulation in the context of combined heat and drought stress. Furthermore, large fractions of the studies reporting on the effects of melatonin in relation to abiotic stress are greenhouse studies with some variation in the environmental parameters. The aim of this study was to investigate the effect of melatonin on combined heat and drought stress at both moderate and severe stress levels during the progression of stress. The study was conducted in fully controlled climate chambers, focusing on the possible mechanistic interactions of the ABA and ROS regulation of stomata. We hypothesized that exogenous melatonin could alleviate negative effects of combined heat and drought stress through an increase in g_s_ mediated by a decrease in ABA. We also hypothesized this mechanism to be mediated through interactions with changes in the enzymatic systems for ROS metabolism.

## 2. Results

### 2.1. Uptake of Melatonin

The leaf melatonin contents were determined to evaluate the effect of the melatonin treatment applied with irrigation. While leaf melatonin was below the detection limits in the non-treated control plants, melatonin was detected in levels of ~600–2000 pmol g DW^−1^ in leaves of plants treated with irritation water containing melatonin (Table 1). This clearly demonstrates that the melatonin supplied to the plants with the irrigation was taken up and transported to the leaves, and thus that the mode of melatonin treatment used here was effective at increasing the melatonin levels in leaves. The four stress treatments showed no significant effects on the melatonin treatment within the moderate stress level at 25 DAS (SRWC ~50% and heat stress for 1) or after progression to severe stress level at 27 DAS (SRWC ~20% and heat stress for 3 days), although it must be noted that the melatonin content had a relatively high degree of variance. Significant differences were only observed between the lower levels in the control and drought treatment at moderate stress level (25 DAS) and higher levels in the drought treatment at severe stress level (27 DAS). This was accompanied by non-significant trends for the melatonin levels, which increased in most stress treatments from moderate stress at 25 DAS to severe stress at 27 DAS, which could indicate that the melatonin was applied at greater rates than it was catabolized in the leaves. 

### 2.2. Physiological Stress Indicators of Heat and Drought Stress—LRWC and F_v_/F_m_

At moderate stress (25 DAS), the leaf relative water content (LRWC) was significantly (*p* < 0.001) affected by heat. An increase was seen in the LRCW from levels in the range of 75–77.5% for plants exposed to 26 °C to a range of 80.5–83% for plants exposed to 38 °C (Table 2). The F_v_/F_m_ was significantly affected by heat and the interaction of heat and melatonin as seen in the decrease in F_v_/F_m_ in treatments at 38 °C compared to 26 °C. This indicated a heat-induced damage to photosystem II after exposure to heat stress for one day at this stress level (Table 2). This pattern did not appear at severe stress (27 DAS), where the LRWC was significantly (*p* < 0.05) affected by drought, where it was seen as a systematic pattern of all drought treatments to have lower values than the irrigated corresponding treatment at the same temperature. This was observed at both 26 °C and 38 °C and for melatonin-treated and untreated plants (Table 2). At severe stress, F_v_/F_m_ was significantly affected by drought and the interaction of drought and heat. A significant decrease from 0.76 to 0.70 was observed in the drought-treated plants compared to irrigated plants at 26 °C in plants not treated with melatonin, indicating that the drought levels at this stress level negatively affected photosystem II (Table 2). Interestingly, the significant differences between treatments at 26 °C and 38 °C observed at moderate stress levels at 25 DAS did not seem present at severe stress levels at 27 DAS. This could indicate a recovery in photosystem II going from day one of heat stress at 25 DAS to day three of heat stress at 27 DAS.

### 2.3. Stomatal Development and Regulation

To evaluate the effects of melatonin and stress treatments on stomatal regulation_,_ we measured the g_s_ and leaf temperature along with anatomical parameters of stomata at moderate and severe stress levels. At moderate stress (25 DAS), the heat stress factor had a significant increasing effect (*p* < 0.001) on the g_s_, whereas the drought level at an SRWC of 50% did not seem to affect the g_s_ significantly at either 26 °C or 38 °C (Figure 1A). The observed increased g_s_ at high temperature is proposed to be mediated by an increase in the stomatal pore area (Figure 2A) rather than the stomatal density (Figure 2E), as the pore area showed a significant increase matching the tendencies in g_s_ as a response to heat.

At severe stress (27 DAS), both the heat and the drought stress factor had significant effects on the g_s_. The drought level of 20% SRWC decreased the g_s_ at both 26 °C and 38 °C compared to the irrigated treatments at these temperatures, whereas higher temperatures seemed to increase g_s_ (Figure 1B). Both effects were mediated by the regulation of the pore area (Figure 2). A significant effect of the melatonin treatment on the g_s_ was found at both moderate (*p* < 0.05) and severe stress (*p* < 0.001).

We observed a general tendency for the melatonin treatments to increase the g_s_ at both moderate and severe stress (Figure 1A,B), although the only significant difference found in the multiple comparisons was between treated and non-treated plants in the combined stress treatment at severe stress. The effects of melatonin on increasing g_s_, and thereby the evaporative leaf cooling, was reflected in the measurements of the leaf temperature, which significantly decreased in the melatonin-treated plants compared to non-treated plants in the combined stress treatment at both moderate and severe stress (Figure 1C,D). The increase in pore area observed as an effect of heat at both 25 and 27 DAS seemed to be explained by an opening of stomata as reflected in the increased stomata width/length ratio (Figure 2C), whereas the length of the stomata was not affected much by treatments (Appendix A). The patterns in the stomatal anatomy on the abaxial leaf side (Figure 2) matched the pattern on the adaxial side (Appendix A), although the pore area and stomatal density were generally lower on the adaxial side than the abaxial.

### 2.4. Hormonal Regulators of Stomata: ABA and ABA-GE

To investigate the hormonal regulation of stomata, the leaf contents of ABA and its conjugated form were quantified at moderate and severe stress levels. Across treatments, ABA and ABA-GE were detected in the range of 2500–6500 pmol/g DW and 1000–4500 pmol/g DW, respectively. No significant effects were observed in the ABA level at moderate stress, signifying that the heat and drought stress at this level did not affect the ABA concentration (Figure 3A). In addition to this, we observed that the patterns in the stomatal aperture at moderate stress, as measured by g_s_ (Figure 1A) and the stomatal pore area (Figure 2A) at moderate stress, were not reflected in their corresponding patterns in the ABA levels. The stomatal effects of heat and melatonin at this stress level was thus not coordinated with the ABA regulation. 

At severe stress, both heat and drought showed significant effects on the ABA levels, with drought increasing the ABA levels at both 26 °C and 38 °C (Figure 3B). Furthermore, ABA levels showed a slight decrease in response to heat stress at both irrigated and drought treatments after three days of severe heat stress (Figure 3B), while a similar effect of heat was not observed at moderate stress after one day of heat stress (Figure 3A). This could suggest a duration-dependent effect of heat stress on the ABA levels. Additionally, a significant increase in ABA-GE was observed as a response to heat at both moderate and severe stress levels (Figure 3C,D), indicating a specific heat-induced response on the conjugation of ABA to glucose to form ABA-GE. However, the increase in ABA-GE did not seem to be followed by a corresponding proportional molar reduction in the ABA level. This signifies that the ABA levels were also governed by either the rate of transport, synthesis, or breakdown rather than regulation by conjugation to glucose alone. The statistical analysis showed no effects of melatonin on the ABA and ABA-GE levels.

### 2.5. Activity of Enzymatic ROS Scavengers

The melatonin treatment showed significant effects on both the SOD and APX activity, observed as a general pattern of an increased activity in melatonin-treated plants compared to non-treated plants (Figure 4A,B,E,F). The activity of SOD was significantly affected by heat at the severe stress level (Figure 4B), while the activity of APX was significantly affected by heat and the interaction of heat and drought (Figure 4F). This was seen as a decrease in SOD in heat treatments and an increase in APX in combined stress treatments at the severe stress level. Adversely, melatonin showed no significant effect on CAT activity, which was significantly affected by the heat treatment (Figure 4C,D). The activity of GR was significantly affected by the melatonin treatment at the severe stress level only, observed as a tendency for the GR activity to decrease as a response to the melatonin treatment. However, the GR activity at severe stress seemed to be triggered by stronger responses to both the heat stress and the interaction of heat and drought.

## 3. Discussion

### 3.1. Melatonin Increases g_s_ through Mechanisms Not Mediated by ABA Regulation

We hypothesized that melatonin would increase g_s_ in combined heat and drought stress mediated by effects on the ABA regulation through effects on ROS signaling. Our results demonstrate a decrease in g_s_ in response to drought and an increase in g_s_ toward heat due to changes in the stomatal pore area. We showed that the heat stress had a predominant effect on the stomatal aperture when the drought level was held constant at an SRWC of 50%, whereas the drought effect to close stomata was more pronounced when the SRWC was lowered to a level of 20%. 

Our results suggest that melatonin increases g_s_ compared to untreated plants in similar stress treatments. This effect was a general trend across stress treatments and was pronounced in the combined stress treatment, where the increase in g_s_ in melatonin-treated plants resulted in better leaf cooling and a significantly lower leaf temperature than non-treated plants at both moderate and severe stress levels. This effect of melatonin did not, however, seem to be caused by effects on the ABA regulation of stomata as hypothesized, as melatonin did not show effects on the ABA levels. In previous studies, exogenous melatonin has shown effects on the expression of genes involved in both the synthesis and breakdown of ABA, causing lower levels of ABA and higher g_s_ upon drought stress in two species of *Malus* [38]. Likewise, melatonin has been shown to alleviate heat-induced leaf senescence in perennial ryegrass by a reduction in the ABA content through the downregulation of transcripts of the synthetic pathway [39]. However, melatonin has also been found to stimulate the accumulation of ABA in wild ryegrass during cold stress [40]. This, together with our results, suggests that melatonin does not directly influence the ABA levels but might interact with colliding signaling pathways depending on the type of stress or combination of stress applied.

It is possible that the effects of melatonin on g_s_ observed here could instead be associated to effects on the enzymatic ROS metabolism. Our results indicate that melatonin increases the activity of APX and SOD. Different patterns of the upregulation of enzymatic ROS scavengers as a response to exogenous melatonin have been reported across species in different stress environments, suggesting that responses are both species and stress-dependent [41]. Several studies even showed a dose dependency in the response of ROS scavengers to exogenous melatonin [42,43,44]. On the cellular level, SOD catalyzes the dismutation of the superoxide radical to form H_2_O_2_, which, in turn, could be degraded by peroxidases, CAT or APX [45]. The increase in SOD and APX activity observed in the melatonin-treated plants could lead to lower levels of superoxide radical and H_2_O_2_. A previous study in tomato has shown a close relationship between the activity of APX and the H_2_O_2_ level in leaves, underlining the importance of this enzyme in ROS removal in tomato [46]. H_2_O_2_ is considered as the main ROS in oxidative burst signaling due to its permeability across membranes and relatively long half-life [47]. H_2_O_2_ is directly involved in the signaling pathway, leading to stomatal closure during stress [16,48]. The increase in g_s_ as a response to melatonin observed in our study could possibly be explained by an inhibition of ROS-signaled stomatal closing by melatonin-induced increases in SOD and APX (Figure 5). 

### 3.2. Heat Stress Decreases ABA Levels, Increases ABA-GE Levels and Opens Stomata

Studies of the effect of heat on stomatal regulation are often complicated by the fact that increasing temperatures have concurrent effects on the VPD, especially if the air humidity cannot be adjusted to ensure an even VPD across different temperatures [49]. As VPD itself has effects on stomatal regulation that differ from temperature [50,51], it is often difficult to decide whether stomatal effects in heat stress are, in fact, a temperature or VPD effect. Heat stress in combination with drought stress is complicated by the fact that a higher temperature increases the transpiration rate to promote evaporative leaf cooling. This might, in turn, result in combined heat and drought-stressed plants ending up being drought-stressed to a larger extent than plants subjected to drought stress alone in experimental setups where the soil water content is not tightly controlled by the use of lysimeters. In the experimental setup of our study, we managed to keep an even VPD at 26 °C and 38 °C by the use of climate chambers with air humidification and a high level of air recirculation. The SRWC was kept constant using lysimeters during the stress period. Hence, the effect of heat was not confounded by differences in VPD between the heat treatment and the control. Likewise, the effects of heat in the combined stress treatment were not confounded by different levels of soil humidity compared to the treatment with drought stress at a moderate temperature. This is a great strength in this study as it yields valuable insights into the effects of heat on stomatal regulation in combined stress. Our results showed a significant effect of heat on increasing the g_s_. We observed a predominant effect of drought over heat on g_s_ at severe drought, whereas the heat effect on stomata seemed to dominate over drought at the moderate drought stress level. This underlines the importance of either controlling the drought level or reporting specific measures on the actual drought level in treatments when studies on combined heat and drought stress are made. 

The temperature in the treatments seemed to be coordinated with effects in the ABA regulation, as a clear increase in ABA-GE was observed as a response to heat at both moderate and severe stress levels. When ABA is conjugated to glucose in ABA-GE, it is considered as physiologically inactive in the stomatal regulation. ABA-GE can be hydrolyzed by the enzymes AtBG1 to release free ABA for the leaf to rapidly adjust to changing environments [52]. Both cold and heat stress have been shown to promote transcriptional responses for genes involved in the conjugation and deconjugation of ABA [53]. Furthermore, the deconjugation of ABA-GE has been demonstrated as an important pathway for enhancing ABA levels in drought stress [54]. To our knowledge, our study is the first to report an increase in ABA-GE as response to high temperatures. Considering that heat waves in nature often instigate drought events by driving an increased evapotranspiration, resulting in the soil drying and an increased drought severity [55], it seems physiologically sound that high temperatures would trigger the leaves to accumulate ABA and store it in the conjugated form to prepare for an upcoming drought event. 

We observed a decrease in ABA due to heat stress. This effect was only present at the severe stress level after three days of heat stress. The decrease in ABA did not seem to be mediated solely by a proportional increase in ABA-GE. Hence, it is likely to be partly explained by changes in the transport or rates of ABA synthesis/breakdown. It is well established that ABA signaling is involved in mediating adaptions to heat stress, such as the expression of heat-shock proteins, thermomorphogenesis and metabolism of molecular antioxidants [56]. An increase in ABA was observed earlier in rice undergoing heat stress in the range of 30–45 °C [57] and cucumber stressed at 40 °C [58]. However, considering that heat stress was not confounded by the effects of VPD or drought severity in our study, we suggest that decreases in ABA could be a possible mechanism involved in the regulation of stomatal opening in heat stress (Figure 5). As our results underline that the duration of heat stress play a role in this effect, additional studies investigating the dependency of duration on heat stress are suggested to explore this possible involvement of ABA regulation on the stomatal opening in heat. 

## 4. Materials and Methods

### 4.1. Plant Material and Unstressed Growing Conditions

Tomato seeds (*S. lycopersicum* L. cv ‘Black Cherry’, Bingenheimer, Echzell, Germany) were sown individually in a peat-based substrate (Pindstrup 2, Pindstrup Mosebrug, Ryomgaard, Denmark) in 11 cm pots and germinated under plastic sheet cover under greenhouse conditions. At 11 days after sowing (DAS), after unfolding of the cotyledons and before formation of the first true leaf, 48 uniform plants were selected out of a total of 80 sown plants and moved to two separate PSI Fytoscope FS-WI walk-in climate chambers (Photon Systems Instruments, Drasov, Czech Republic). In the chambers, plants were distributed on four 12-scale DroughtSpotter lysimeter units (Phenospex, Heerlen, The Netherlands). The DroughtSpotters were operated in hold mode with irrigation to the defined target weight at every second hour before the onset of stress treatments and every half hour during the stress period. For the growth at unstressed conditions, the seedlings were maintained at 410 g ramping up to 420 g according to the growth of the plants, corresponding to a soil relative water content (SRWC) of 90%. Temperature was set at 26 °C/18 °C and relative humidity 61%/59% day/night with a 14 h photoperiod. Grow light was supplied by build-in broad band white LED panels at photosynthetic photon flux density (PPFD) of 440 µmol m^−2^ s^−1^ measured at the position of the top leaf of the plants at the start of the experiment, which increased to 450 µmol m^−2^ s^−1^ at the end of the experiment as the plants grew toward the lamps. The photoperiod was staggered with two hours between the two chambers to be able to perform measurements at the same time with respect to the start of the photoperiod in both chambers. The CO_2_ concentration was maintained at 410 ppm for the entire experiment.

### 4.2. Melatonin Treatment

The melatonin treatment was applied with the irrigation water at a concentration of 150 µM from day 11, when the plants were transferred to the growth chambers and to the onset of stress treatments at day 22. To prepare the solution, melatonin (Sigma-Aldrich, Søborg, Denmark) was dissolved in ethanol to a concentration of 20 mg/mL, which was diluted into the irrigation water containing nutrients (N 190 ppm, P 35 ppm, K 275 ppm, Mg 40 ppm, Ca 170 ppm, Na 18 ppm, Cl 58 ppm, SO_4_ 94 ppm, Fe 1.5 ppm, Mn 1 ppm, B 0.3 ppm, Cu 0.15 ppm, Zn 0.20 ppm, Mo 0.05 ppm, Si 12 ppm). For the negative control to the melatonin irrigation water, an equal amount of pure ethanol was diluted into the same nutrient solution as described above. As melatonin is reported to be unstable, the irrigation solutions were replaced every second day. The melatonin concentration in the irrigation tank was monitored twice a day by taking samples of the irrigation water at the entrance valve of the DroughtSpotter tables and obtaining a UV absorbance spectrum using a UV-1700 spectrophotometer (Shimadzu, Ballerup, Denmark). From the onset of the drought, heat and combined stress treatments, the irrigation system of the DroughtSpotters was operated with only clean nutrient solution, and melatonin was supplied manually to avoid differences in melatonin dosage caused by differences in water use between treatments. This was carried out by adding 5 mL of 1.05 mM melatonin in irrigation water to each pot of the melatonin treatments once a day, corresponding to the same daily dose of melatonin that the plants were supplied with by the irrigation before the onset of treatments. Negative controls of the melatonin treatment were supplied with 5 mL nutrient solution containing the same amount of ethanol as the melatonin treatment. 

### 4.3. Stress Treatments

At 22 DAS, plants from the melatonin and control treatment were divided into four treatments: no stress (control), individual drought stress, individual heat stress and combined heat and drought stress. The stress treatments are depicted in Figure 6. The drought treatment was applied in two phases: moderate drought (soil relative water content (SRWC) = 50%) and severe drought (SRWC = 20%). First, the SRWC was ramped down gradually over 2 days to target weight of 360 g corresponding to an SRWC of 65%. When this was reached at the start of the day at 24 DAS, the heat treatment was initiated in the individual heat and combined stress treatments and the target weight for moderate drought stress was set at 320 g (SRWC = 50%). Physiological measurements and samples for moderate drought stress were collected after this target weight was reached for all plants in the drought and combined stress treatment, taking place at midday at 25 DAS. Hereafter, the target weight of the drought treatments was reduced to 250 g, corresponding to an SRWC of 20%. Physiological measurements and samples at the more severe drought conditions were made when this target weight was reached at 27 DAS. Plants of the control treatment and the individual heat treatment were maintained at SRWC of 90% throughout the whole experiment.

The climate conditions for the control and individual drought-stressed plants were kept at 26 °C/18 °C and RH:61%/59%; VPD: 1.31/0.84 day/night. For the heat stress and combined heat and drought stress treatment, the settings were 38 °C/30 °C; RH:80%/80%; VPD: 1.31/0.84 day/night. 

### 4.4. Chlorophyll Fluorescence

F_v_/F_m_ was measured six hours into the photoperiod. A Mini-PAM fluorometer (Walz GmbH, Effeltrich, Germany) was used to assess the maximum quantum efficiency of PSII photochemistry (F_v_/F_m_). Measurements were taken on three fully unfolded light-exposed leaflets for three plants of each treatment after dark adaption by leaf clips for 30 min.

### 4.5. g_s_ and Leaf Temperature

g_s_ and leaf temperature were measured in situ at 6–7 h into the photoperiod using an LI-600 leaf porometer (LI-COR Biosciences, Cambridge, UK) operated in auto gsw mode. The instrument was tempered to the temperature in the chambers for approximately one hour prior to measurements to avoid effects of condensation in the instrument. For each treatment, 18 individual measurements on the abaxial leaf side of the two uppermost fully expanded leaves were taken. All measurement were performed in mixed sequence to avoid systematic errors between treatments caused by temporary environmental changes or plant circadian rhythm.

### 4.6. Stomatal Imprints

Stomatal anatomy and density from both abaxial and adaxial surface of the leaf were assessed by taking imprints from the first primary leaflet from the second uppermost fully unfolded leaf on both sides of the leaves in three biological replicates per treatment using the silicon rubber impression technique [59]. Imprints were made using impression material (elite HD+, Zhermack, Badia Polesine, Italy), and imprints were transferred to microscopic slides using nail varnish on the original imprints. Images were acquired at six different locations for each imprint using a Nikon AZ100 microscope (Nikon Corp., Tokyo, Japan) equipped with a Nikon DS-Fi1 camera. Images were analyzed using ImageJ software (version 1.51). For each picture, the stomata number was counted to determine the stomatal density. For measurements of stomatal anatomy, stomatal length, stomatal width, pore length and pore width were determined for all stomata in a randomly selected quarter of three images per biological replicate. The pore area was estimated assuming the shape of an ellipse, and stomata size was estimated by the area of the rectangle encasing the stomata.

### 4.7. Leaf Relative Water Content (LRWC)

Samples for determination of the LRWC were obtained seven hours into the photoperiod. One leaf disc (22 mm diameter) was punched from a leaflet in three biological replicates per treatment and weighed using an analytical weighing scale to determine FW. The leaf discs were soaked in demineralized water in petri dishes overnight and blotted with tissue paper before determining the turgid weight (TW). Finally, leaf discs were dried at 80 °C for 24 h to determine the DW. LRWC was calculated as: LRWC% = (FW − DW)/(TW − DW).

### 4.8. LC-MS/MS for ABA, ABA-GE and Melatonin

Samples for LC-MS/MS were harvested seven hours into the photoperiod, froze in liquid nitrogen and lyophilized. Approximately 7 mg of ground, lyophilized leaf material was weighed into 2 mL sample tubes and added with 700 µL cold extraction solvent of methanol/water/acetic acid (10/89/1, *v*/*v*). Then, they were sonicated in ice bath for 15 min and then mixed at 1400 rpm at 4 °C for 1.5 h using an Eppendorf thermomixer. Plant debris was removed from the samples by centrifuging at 20,800× *g* at 4 °C for 10 min. Hereafter, the supernatant was filtered through 0.22 µ, nylon membrane Costar Spin-X Centrifuge filters (Corning Incorporated, Corning, NY, USA) by spinning at 10,000 g for five minutes at 4 °C. Filtrated samples were transferred to glass vials and kept at −20 °C until analysis. Extractions were prepared in analytical duplicates within three biological replicates per treatment. 

Targeted analysis of ABA, ABA-GE and melatonin was performed using an Agilent 6495 triple quadrupole mass spectrometer (Agilent Technologies, Santa Clara, CA, USA) equipped with Agilent Jet Stream Electrospray Ionization (ESI) source, coupled to a 1290 infinity II LC system (Agilent Technologies). Chromatographic separation was performed on an Agilent ZORBAX RRHD Eclipse Plus 95 Å C18, 2.1 × 100 mm, 1.8 μm column (Agilent Technologies). Gradient elution was performed with H_2_O + 0.1% formic acid (solvent A) and acetonitrile + 0.1% formic acid (solvent B) at a constant flowrate of 0.4 mL min^−1^ and sample injection volume of 1 µL. A gradient profile with the following proportions of solvent B was applied (t(min),%B): (0, 20), (1, 20), (8, 90), (9, 90), (10, 20), (12, 20), with column equilibration between each run. The analyses were performed using electrospray ionization in negative ion mode for ABA and ABA-GE and positive mode for melatonin. Capillary voltage was 3000 V and 3500 V for negative and positive mode, respectively, and charging voltage was 1500 V for both negative and positive mode. A drying gas flow of 14 L/min at 250 °C, sheath gas flow of 12 L/min at 250°C and nebulizer pressure of 35 psi were used. Acquisition was performed by multiple reaction monitoring (MRM). Fragmentor voltage at 380 V and cell accelerator voltage at 5 V were used in all cases. ABA and ABA-GE were detected with a dwell time of 100 ms, whereas 200 ms was used for melatonin. Optimized settings of collision energy and transitions used for the individual compounds were as shown in Table 3. Calibration curves of external standards of ABA (Sigma-Aldrich), ABA-GE (OlChemIm Ltd., Olomouc, Czech Republic) and melatonin (Sigma-Aldrich) were used for quantification. For quality assurance of the reproducibility of the MS detection and the stability of sample standing in the auto sampler, the same mixed sample of extract was run for every 20th sample in the sequence.

### 4.9. Assays of Antioxidant Enzymes (SOD, CAT, APX, GR)

Assays for antioxidant enzymes SOD, CAT, APX and GR were performed based on Elavarthi and Martin [60], modified and optimized for analysis in 96-well microplates [61]. Sample extracts were prepared by grinding leaf tissue in liquid nitrogen using a mortar and pestle. Each biological replicate was separated into three technical replicates by transferring 100 mg leaf tissue into 2 mL sample tubes. Enzymes were extracted in 1.2 mL of 0.1 M potassium phosphate buffer with 0.1 mM EDTA (pH 7.8) by mixing for 15 min at 4 °C using an Eppendorf thermomixer, spun down at 5000 g for 5 min and then re-extracted in 0.8 mL. Combined supernatants were stored on ice until assay reactions. All enzyme assays were evaluated using a Synergy 2 plate reader (Biotek Instruments, Winooski, VA, USA).

The SOD activity was evaluated by inhibition of the reduction of p-nitro-blue tetrazolium chloride (NBT). The reaction mixture containing 50 mM phosphate buffer (pH 7.8), 2 mM EDTA, 9.9 mM L-Methionine, 55 µM NBT, 0.025% Triton-X, 40 µM riboflavin and 20 µL sample extract was mixed at a total volume of 250 µL in the microplate (UV-VIS, 96/F, Eppendorf AG, Hamburg, Germany). Sample blanks were prepared using extraction buffer instead of leaf extract. The microplate was placed in a plate shaker and reactions were initiated by exposing the plate to a light intensity of 100 µmol m^−1^ s^−1^ supplied by white LEDs for four minutes. Hereafter, the reaction was stopped by transferring the plate on ice in darkness, and absorbance at 560 nm was read immediately. One unit of SOD was defined as the amount of enzyme that produced a 50% inhibition of NBT reduction, using the illuminated sample blank as the 0% reference and sample blank kept in darkness as the 100% reference. 

CAT activity was determined by the decomposition of H_2_O_2_ followed as a decrease in absorbance at 240 nm. The reaction mixture contained 50 mM phosphate buffer (pH 7.0), 15 mM H_2_O_2_ and 20 µL extract in a total volume of 250 µL per well. The absorbance at 240 nm was measured continuously for 5 min, and the reaction rate was determined by linear fitting to the decrease in absorption for each reaction well. The extinction coefficient of H_2_O_2_ (43.6 M^−1^ cm^−1^) was used to express the enzyme activity in terms of mmol of H_2_O_2_ min^−1^ gFW^−1^.

The activity of APX was determined by the decrease in absorbance at 290 nm due to oxidation of ascorbate. The reaction mixture contained 50 mM phosphate buffer (pH = 7.0), 0.5 mM ascorbate, 15 µL extract and 0.5 mM H_2_O_2_ in a total volume of 250 µL. The reactions were initiated by the addition of H_2_O_2_ and absorbance at 240 nm was measured continuously for 10 min. The APX activity was expressed in units of mmol of oxidized ascorbate min^−1^ gFW^−1^ using the extinction coefficient of reduced ascorbate (2.8 mM^−1^ cm^−1^).

The GR activity was determined by following the reduction of 5,5-dithio-bis-(2-nitrobenzoic acid) (DTNB) to 2-nitro-5-thiobenzoic acid (TNB) by GSH formed by the GR-catalyzed reduction of GSSG. The reaction mixture contained 50 mM phosphate buffer (pH 7.8), 0.75 mM DTNB, 0.1 mM NADPH, 15 µL extract and 1 mM GSSG in a total volume of 250 µL per well. The reaction was initiated by the addition of GSSG, and the reaction rate was monitored by following the formation of TNB by measuring absorbance at 412 nm continuously over 5 min. GR activity was expressed in units of mmol of TNB min^−1^ gFW^−1^ using the extinction coefficient for TNB (14.15 M^−1^ cm^−1^).

### 4.10. Statistics

Three-way ANOVA was performed to determine the possible interaction between the three treatment factors (melatonin, heat and drought) followed by post hoc Tukey test for multiple comparisons between treatments using GraphPad Prism (version 9.4.1, GraphPad, San Diego, CA, USA). Statistical analyses were performed within the two levels of stress separately for all data except for LC-MS/MS data for melatonin level, which was performed across the two stress levels, including only data from melatonin-treated plants. 

## 5. Conclusions

The results of our study indicate that the exogenous application of melatonin can modulate the stomatal behavior in combined heat and drought stress. This did not seem to be mediated through effects on the ABA regulation as hypothesized. The effect might rather be mediated through impacts on the ROS metabolism. The study has provided novel insights into the possible involvement of ABA metabolism in high-temperature-induced stomatal opening not related to exogenous melatonin. 

## Figures and Tables

**Figure 1 plants-12-01156-f001:**
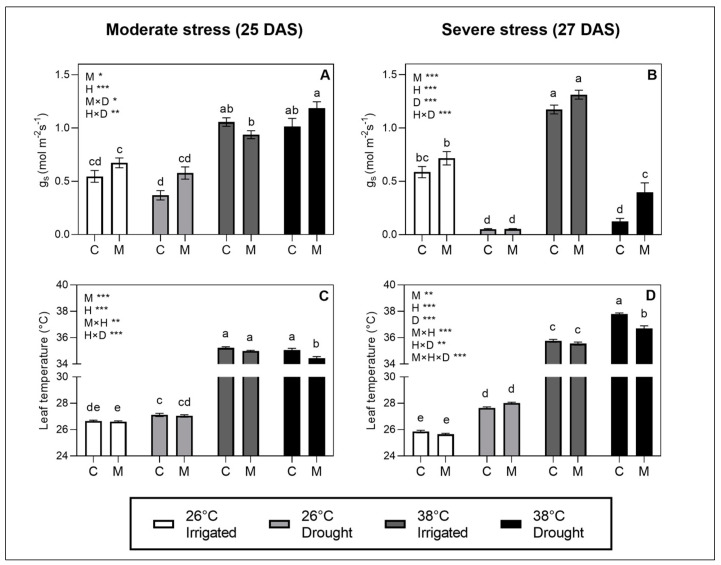
Stomatal conductance (g_s_) (**A**,**B**) and leaf temperature (**C**,**D**) measured in situ on plants during moderate stress (drought-stressed at SRWC ~50% and heat stress for 1 day) and severe stress (drought-stressed at SRWC ~20% and heat stress for 3 days). Values are mean ± SE. The effects of the three factors studied—heat stress (H), drought stress (D), melatonin treatment (M)—and their interaction are indicated in the left, upper corners of graphs. Significance levels are indicated by ***, *p* < 0.001; **, *p* < 0.01; *, *p* < 0.05, whereas factors with no significant effect (*p* > 0.05) are not listed. Lower-case letters on bars indicate differences between groups within each stress level significant at *p* < 0.05 in the post hoc multiple comparisons.

**Figure 2 plants-12-01156-f002:**
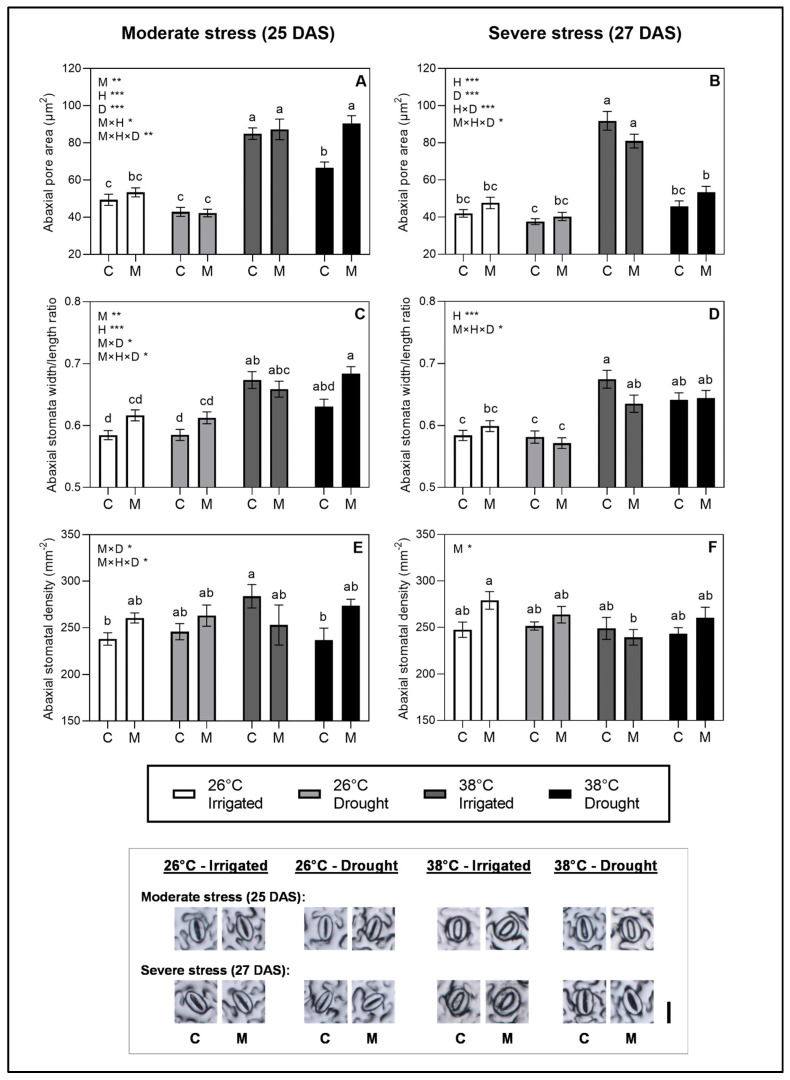
Stomatal anatomy from the abaxial leaf side on plants during moderate stress (drought-stressed at SRWC ~50% and heat stress for 1 day) and severe stress (drought-stressed at SRWC ~20% and heat stress for 3 days). Stomatal anatomy included abaxial pore area (**A**,**B**), abaxial stomata width/length ratio (**C**,**D**) and abaxial stomatal density (**E**,**F**) during moderate and severe stress. Values are mean ± SE. The effects of the three factors studied—heat stress (H), drought stress (D), melatonin treatment (M)—and their interaction are indicated in the left, upper corners of graphs. Significance levels are indicated by ***, *p* < 0.001; **, *p* < 0.01; *, *p* < 0.05, whereas factors with no significant effect (*p* > 0.05) are not listed. Lower-case letters on bars indicate differences between groups within each stress level significant at *p* < 0.05 in the post hoc multiple comparisons. Representative images of stomata from each treatment at each stress level are shown in subfigure below graphs. Scale bar (lower right corner), 25 µm.

**Figure 3 plants-12-01156-f003:**
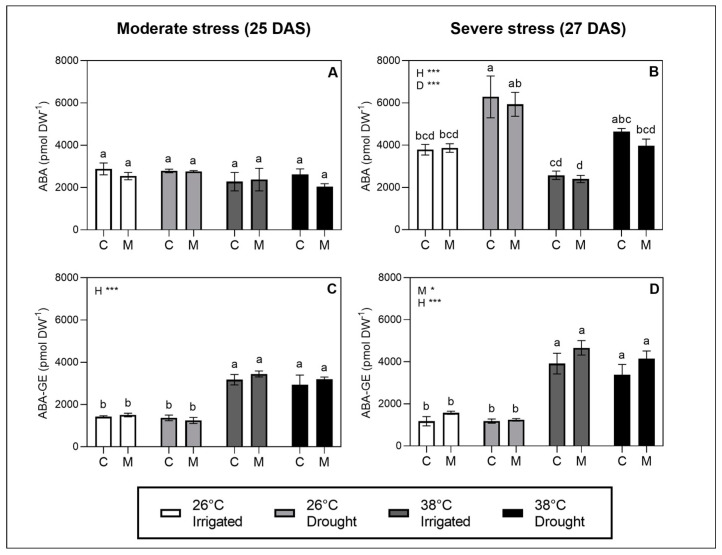
LC–MS analysis of ABA (**A**,**B**) and ABA-GE (**C**,**D**) performed on leaf material harvested during moderate stress (drought-stressed at SRWC ~50% and heat stress for 1 day) and severe stress (drought-stressed at SRWC ~20% and heat stress for 3 days). Values are mean ± SE. The effects of the three factors studied—heat stress (H), drought stress (D), melatonin treatment (M)—and their interaction are indicated in the left, upper corners of graphs. Significance levels are indicated by ***, *p* < 0.001; *, *p* < 0.05, whereas factors with no significant effect (*p* > 0.05) are not listed. Lower-case letters on bars indicate differences between groups within each stress level significant at *p* < 0.05 in the post hoc multiple comparisons.

**Figure 4 plants-12-01156-f004:**
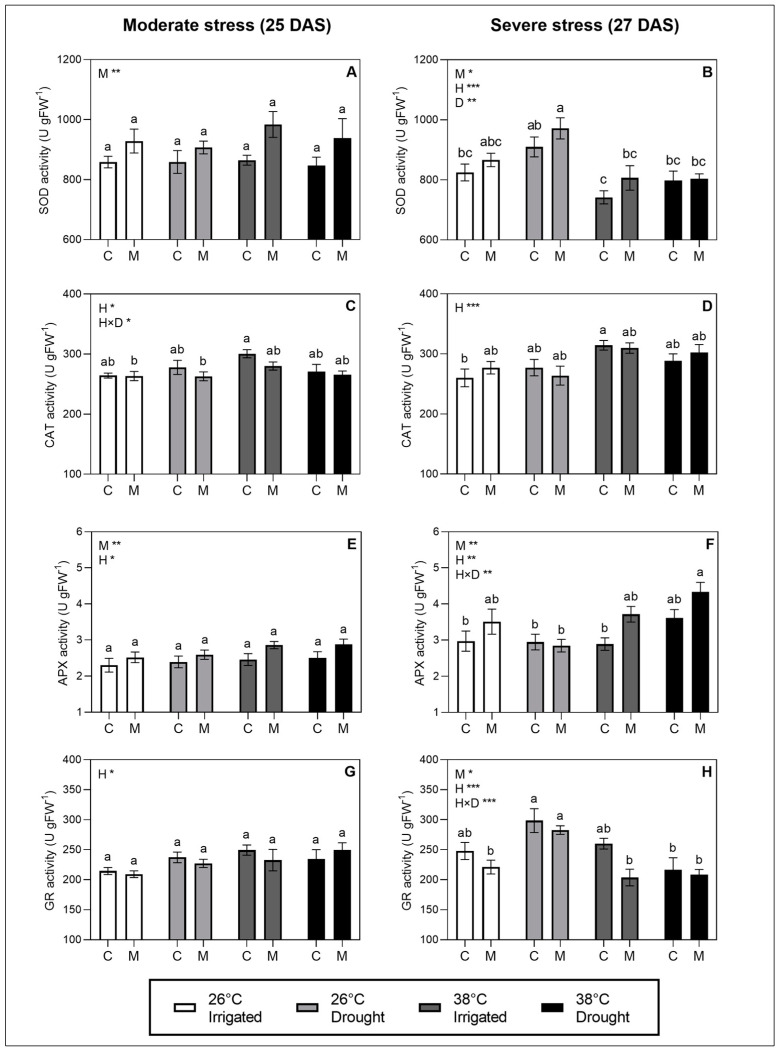
Enzymes assay on fresh leaf material for SOD (**A**,**B**), CAT (**C**,**D**), APX (**E**,**F**) and GR (**G**,**H**) at moderate stress (drought-stressed at SRWC ~50% and heat stress for 1 day) and severe stress (drought-stressed at SRWC ~20% and heat stress for 3 days). Values are mean ± SE. The effects of the three factors studied—heat stress (H), drought stress (D), melatonin treatment (M)—and their interaction are indicated in the left, upper corners of graphs. Significance levels are indicated by ***, *p* < 0.001; **, *p* < 0.01; *, *p* < 0.05, whereas factors with no significant effect (*p* > 0.05) are not listed. Lower-case letters on bars indicate differences between groups within each stress level significant at *p* < 0.05 in the post hoc multiple comparisons.

**Figure 5 plants-12-01156-f005:**
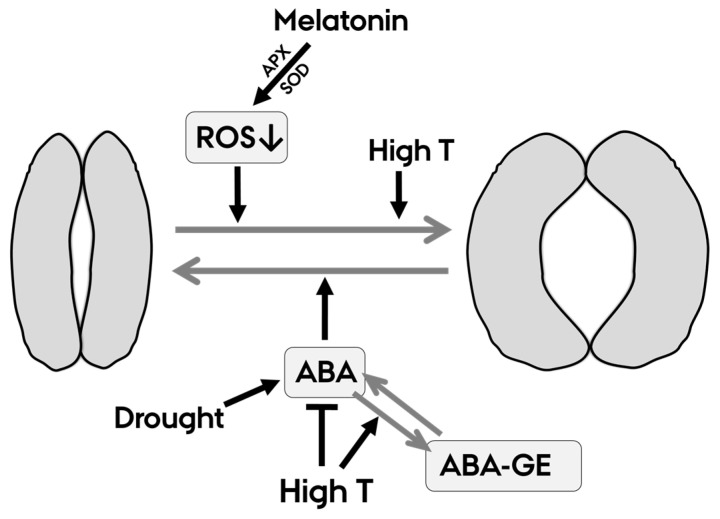
Graphic summary of the proposed mechanisms interacting in stomatal regulation by melatonin, heat and drought at combined stress. Upper part of figure: stomatal opening is promoted separately by both heat effects of stomata and by inhibition of ROS signaling in stomatal closing induced by melatonin. Lower part: stomatal closing is promoted by an ABA accumulation as response to drought. Our results suggest that heat counteracts the accumulation of ABA by increasing the conjugation to ABA-GE and through additional effects on either transport, synthesis or breakdown.

**Figure 6 plants-12-01156-f006:**
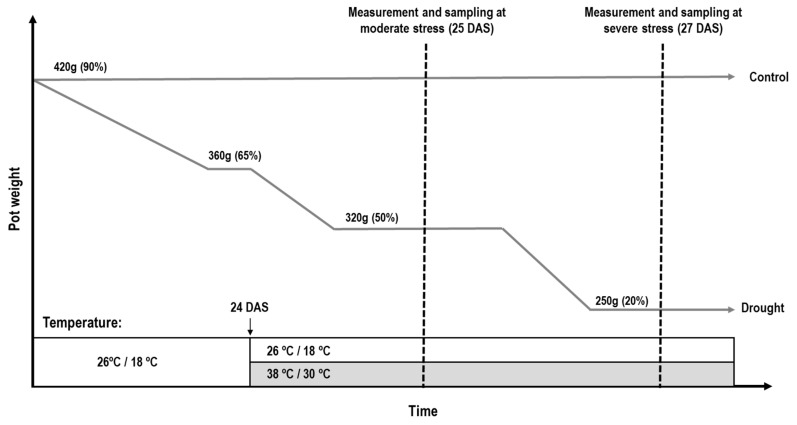
Overview of the workflow for application of drought and heat stress. The 22-day-old tomato plants were divided into either control, individual drought stress, individual heat stress or combined heat and drought stress. Plants of the control treatment were maintained at 420 g (SRWC: 90%) at 26/18 °C throughout the whole experiment. Plants of the individual drought treatment were all ramped down to a set point of 360 g (SRWC: 65%), which was reached at 24 DAS. From here, they were further ramped to a set point of 320 g (SRWC: 50%) for measurements and sampling at moderate drought stress and, hereafter, a set point of 250 g (SRWC: 20%) for measurements at severe drought stress. Plants in the heat treatment were subjected to increased temperature at 38/30 °C from 24 DAS until the end of experiment. The plants of the combined stress treatment received a combination of the drought and heat treatment.

**Table 1 plants-12-01156-t001:** LC–MS analysis of melatonin performed on leaf material harvested during moderate stress (drought-stressed at SRWC ~50% and heat stress for 1 day) and severe stress (drought-stressed at SRWC ~20% and heat stress for 3 days). Values are mean ± SE. Letters indicates differences between stress treatment groups across the two stress levels. ND, not detected.

	26 °C Irrigated	26 °C Drought	38 °C Irrigated	38 °C Drought
**Melatonin (pmol g DW^−1^)**				
**Moderate stress (25 DAS)**				
Control	ND	ND	ND	ND
Melatonin	634.8 ± 209.7 b	600.8 ± 100.9 b	747.0 ± 192.3 ab	1123.2 ± 428.7 ab
**Severe stress (27 DAS)**				
Control	ND	ND	ND	ND
Melatonin	1173.3 ± 468.6 ab	2067.6 ± 300.9 a	1404.8 ± 313.1 ab	1120.9 ± 321.1 ab

**Table 2 plants-12-01156-t002:** Leaf relative water content (LRWC) and PSII photochemistry (F_v_/F_m_) in dark-adapted leaves obtained during moderate stress (drought-stressed at SRWC ~50% and heat stress for 1 day) and severe stress (drought-stressed at SRWC ~20% and heat stress for 3 days). Values are mean ± SE. The effects of the three factors studied—heat stress (H), drought stress (D), melatonin treatment (M)—and their interaction are indicated in the column to the right. Significance levels are indicated by ***, *p* < 0.001; **, *p* < 0.01; *, *p* < 0.05, whereas factors with no significant effect (*p* > 0.05) are not listed. Letters next to the values indicates differences between groups significant at *p* < 0.05 in the post hoc multiple comparisons.

	26 °C Irrigated	26 °C Drought	38 °C I rrigated	38 °C Drought	ANOVA
**LRWC (%)**					
**Moderate stress (25 DAS)**					
Control	77.5 ± 1.1	77.0 ± 1.6	80.4 ± 2.9	82.8 ± 1.9	H ***
Melatonin	76.1 ± 0.9	75.1 ± 0.9	81.6 ± 2.8	82.2 ± 1.2	
**Severe stress (27 DAS)**					
Control	79.3 ± 1.6	75.8 ± 1.8	80.9 ± 4.2	76.5 ± 2.0	D *
Melatonin	77.0 ± 2.0	75.5 ± 0.9	84.5 ± 2.4	78.1 ± 2.3	
**F_v_/F_m_**					
**Moderate stress (25 DAS)**					
Control	0.78 ± 0.004 a	0.78 ± 0.007 a	0.74 ± 0.006 cd	0.73 ± 0.008 d	H *** M × H *
Melatonin	0.76 ± 0.004 ab	0.76 ± 0.004 abc	0.73 ± 0.005 cd	0.74 ± 0.006 bcd	
**Severe stress (27 DAS)**					
Control	0.76 ± 0.011 a	0.70 ± 0.028 b	0.75 ± 0.006 ab	0.76. ± 0.005 a	D ** D × H ***
Melatonin	0.77 ± 0.004 a	0.72 ± 0.014 ab	0.74 ±0.006 ab	0.74 ± 0.005 ab	

**Table 3 plants-12-01156-t003:** Optimized MS/MS settings and retention time with the chromatography used.

Compound	Precursor Ion (*m*/*z*)	Product Ion (*m*/*z*)	Collision Voltage (V)	Retention Time (min)
ABA	263	153	13	3.53
ABA-GE	425	263	18	2.00
Melatonin	233.1	174	15	2.88

## Data Availability

Data will be made available on request.

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
