# Peer review of "Exogenous Melatonin Alters Stomatal Regulation in Tomato Seedlings Subjected to Combined Heat and Drought Stress through Mechanisms Distinct from ABA Signaling"

_plants, 2023, doi:10.3390/plants12051156_

Round 1

Reviewer 1 Report

1, The innovation points in the abstract are not very clear. The title of the article refers to melatonin, but the last sentence of the abstract is the relationship between ABA and stomata, which needs to be improved.

2, L17: Please mark the full name of SRWC when it is first mentioned.

3, L32:The first sentence should be two characters blank. There are many similar problems in this MS. Please check it.

4, L120: Why choose 25 days and 27 days instead of two time points with large time span. Please provide evidence when you first mention it

5, L138: 75%-77.5%; 80.5-83%

6, I think the data in Table 1 and 2 is more concise and clear in the column chart, and the specific data can be used as a supplementary table.

7, Figure 2 should add real pictures of stomatal aperture and number.

8, L475: Please quote the relevant report on measuring stomata in tomato, not Rosa hydrida

9, L481: You should select as many leaves as possible instead of selecting several areas from three biological replicates, which may lead to large errors, unless you will specifically select leaves with similar stomatal size as the final data.

10, L399: Why not spray melatonin directly but load it into nutrient solution?

11, All pictures should be centered for better appearance.

12, L440. Please note whether the unit of ºC is wrong(26ºC/18 ºC)?

Author Response

Dear Editor and reviewers:

We hereby send you the revised manuscript with manuscript ID: plants-2234500. Please note, that the manuscript was previously entitled “Effects of melatonin on stomatal regulation in tomato seedlings subjected to combined heat and drought stress”. However, the title have now been updated to “Exogenous melatonin alters stomatal regulation in tomato seedlings subjected to combined heat and drought stress through mechanisms distinct from ABA signaling”.

We appreciate the thorough review of our manuscript and the very constructive comments and suggestions. We have done our best to respond and consider the reviewers comments. Below we addressed all comments and suggestion from the reviewers. Our responses are in italic.

Respond to reviewer #1

1, The innovation points in the abstract are not very clear. The title of the article refers to melatonin, but the last sentence of the abstract is the relationship between ABA and stomata, which needs to be improved.

Yes, we agree that the last sentence of the abstract does seem a little off topic in context of the title. We thank the reviewer for this observation. We now changed the title to “Exogenous melatonin alters stomatal regulation in tomato seedlings subjected to combined heat and drought stress through mechanisms distinct from ABA signaling” (L2-4). We think that this new title sets a theme that is inclusive to both the findings on melatonin effects on stomata in combined stress and the effects on ABA in high temperature – both stated in the conclusive part of abstract. Moreover, the corresponding section in abstract was changed as follows: ‘The ABA metabolism and conjugation of ABA might play a role behind stomatal opening towards high temperatures. We provide evidence that the melatonin increases gs in combined heat and drought stress, but the effect is not mediated through ABA signaling.’ Please check Line 22-26.

2, L17: Please mark the full name of SRWC when it is first mentioned.

We added the full name of the abbreviation (L18).

3, L32:The first sentence should be two characters blank. There are many similar problems in this MS. Please check it.

We formatted this paragraph to start with an indent as suggested (L32). Similar changes in format have been made other places in the MS (L 85, 368).

4, L120: Why choose 25 days and 27 days instead of two time points with large time span. Please provide evidence when you first mention it

We agree, that the 25 and 27 DAS miss some context, when it is first read it in section 2.1. The scope of our experiment was to study the stress reactions during the progression of stress. We chose the span from 25 and 27 days, because those two days were simply the time needed for the drought stress to progress from SRWC = 50% to SRWC =20% in our experimental setup. This design of the stress treatments should hopefully be clear from the description of the stress treatments in the method section (4.3). We have added some further description on the scope for our experimental design along with the description of the aim (L 114) and rewritten the presentation of the first results under section 2.1 (L129-132) to clarify this aspect earlier in the MS.

These sentences now read:

L110

“The aim of this study was to investigate the effect of melatonin on combined heat and drought stress at both moderate and severe stress level during the progression of stress.”

L125-129

“The four stress treatments showed no significant effects on the melatonin treatment within the moderate stress level at 25 DAS (SRWC ~50% and heat stress for 1) or after progression to severe stress level at 27 DAS (SRWC ~20% and heat stress for 3 days), although […]”

5, L138: 75%-77.5%; 80.5-83%

        This is now corrected

6, I think the data in Table 1 and 2 is more concise and clear in the column chart, and the specific data can be used as a supplementary table.

For table 1 we further have a large fraction of treatments, where the melatonin levels were below the limit of detection. We think that this fraction of “not detected” (ND) values are communicated more clearly in the table format than in a column chart, where they would appear as missing columns.

Table 1 and 2 presents some indicators of the applied treatments (melatonin, heat and drought). This is supportive data of the study, that is not central for the discussion section. The most important data of the study are the data presented in figures (Fig. 1-4), which form the basis of the mechanistic discussion in section 3.1. and 3.2. While reading the discussion, they reader might want to revisit the data in Fig. 1-4, while the data of Table 1-2 in not central for the discussion section. We therefore think that presenting the data contained in table 1 and 2 in the compact table format rather than column charts gives an overall better structure and order of the paper as a whole.

With the reasoning above, we would suggest to keep table 1 and 2 in the format of tables.

7, Figure 2 should add real pictures of stomatal aperture and number.

We agree, that representations of individual stomata could be added to the figure to improve the dissemination of the data. With 8 columns per chart, it is not applicable to place pictures of representative stomata above each column, as seen in some papers with fewer treatments. The pictures would become too small. Instead, we added it in a separate window below the charts with the representative stomata for all treatments.

We do think, that the original Fig 2 with 8 column charts + the extra window with pictures becomes too large for showing all elements in a suitable scale, considering the space needed for header, footer and figure text on the page as well. Therefore we reduced the number of charts in the figure, and moved the stomatal size data to Table S1, where we already had additional stomatal parameters. We think that this prioritization of data is better, since the differences in the stomatal size where mediated through stomatal widening/increase in pore area, which is already reflected in the figure by the trends in pore area and W/L ratio.

Regarding stomatal density/number: it would require uncropped images to show good representations of the stomatal density in tomato leaves (see example below). Each picture would have to be dimensioned to a size, where we could fit 2 pictures within the width of a page and still have a reasonable image resolution. With our 8 treatments at two days, it would require two pages to show a representation of the stomatal density for each treatment. We think that having a figure of that size in the paper would be unproportional large compared to the information that it would communicate. Instead, we added such figure to the supplementary material.  

We have our new Fig. 2 in the MS, updated references to fig 2, updated Table S1 and added figures for representing stomatal density in the supplementary material.

Figure 1Example of full size microscope image exemplifying stomatal density

8, L475: Please quote the relevant report on measuring stomata in tomato, not Rosa hydrida

The previously quoted paper contains a description of the method from when it was implemented in our group. The materials, technical setup and protocol are the same for roses as for tomatoes. We changed the reference to a more recent paper, where we used the technique in tomato:

Zhou R, Yu X, Wen J, Jensen NB, Dos Santos TM, Wu Z, Rosenqvist E, Ottosen CO. 2020. Interactive effects of elevated CO2concentration and combined heat and drought stress on tomato photosynthesis. BMC Plant Biology 20, 1–12.

9, L481: You should select as many leaves as possible instead of selecting several areas from three biological replicates, which may lead to large errors, unless you will specifically select leaves with similar stomatal size as the final data.

We are aware that taking stomatal imprints from many different leaves is a sampling method used within the field of ecology. We used an approach where we took imprints of leaves of similar developmental stage in all treatments (as described in L 489-491), which is the common sampling strategy used for studying short term stress responses to abiotic stress in controlled experimental setups. Heat and drought stress is reported to cause effects on the stomatal size. Therefore, we do not think that filtering of the final data based on stomatal size would be appropriate data handling, considering the aim and scope of this paper.

10, L399: Why not spray melatonin directly but load it into nutrient solution?

Yes, it is a good question, and we had many discussions on this in the experimental design. Exogenous melatonin is applied either by spraying or by irrigation. It’s hard to say which method is mostly used, but both methods are common. Very few papers so far actually report the achieved concentration of melatonin in plant tissue after treatment, which makes it difficult to compare effects across studies, and one of the reasons that we wanted to include analysis of melatonin in this study. As we wanted to have a measure of melatonin in the leaf tissue, treating the plants by irrigation seemed to be the best method of application. Spraying would distribute melatonin on the leaf surface, from where it would have to be absorbed at an unknown rate. In a later chemical analysis of the leaf tissue, it would not be possible to distinguish melatonin that was taken up by the leaves from melatonin that was just deposited on the leaf surfaces from the spraying.

When we treat the plants by irrigation, we only expose the surface of the roots. So when we later measure melatonin in leaves, we know that it was taken up by the plants (and transported to the leaves), and hence present inside the leaf tissue, not just on the surface. Thus, this method of application together with our analysis for melatonin gives us a surer confirmation, that the applied melatonin was actually present in the plants at a cellular level. It also gives a more certain estimation of the active melatonin content in tissue of the leaves.

11, All pictures should be centered for better appearance.

Yes. We corrected this error.

12, L440. Please note whether the unit of ºC is wrong(26ºC/18 ºC)?

        Yes, that was wrong. We corrected that that.

We deeply appreciate your consideration of our manuscript, and we look forward to hearing good news from you.

Best wishes

Dr. Rong Zhou, Associate Professor from NAU

Email: [email protected] / [email protected]

Aarhus University

Agro Food Park 48, Aarhus N, Denmark

Reviewer 2 Report

Dear Authors

Greetings!!!!

I have read your manuscript entitled “Effects of melatonin on stomatal regulation in tomato seedlings subjected to combined heat and drought stress” with full interest and I found that the manuscript is very well written, and all the sections (Introduction, Material Method, Discussion) are described correctly and nicely by the authors, the overall quality of the manuscript is good. So, I do not have many major comments except few. These are as follows.

Ø  I would like to ask authors that have they did any seed viability test? I do not found any information related to seed homogeneity and authenticity in material method section. If they did kindly mention it in material method section.

Ø  Proline content is one of the most important stress markers but unfortunately it is not done by the authors, So I strongly recommend authors to include this parameter in their next study if they planned to work on stress.

Ø  As far as Result section is concerned the way authors explained the results are quite complicated to understand reason being authors were trying to explain and correlate so many parameters in single paragraph that may cause a difficulty for the reader to focus on one particular parameter For example,

Point 2.3 The results of Stomatal development and regulation.

Under this heading authors not only mention the finding of stomatal conductance and leaf temperature but also mentioned the findings of stomatal width, pore area, stomatal width/length ratio and mentioned the correlations among all these parameters. (that are given in two Figures  i.e Fig 1 and Fig 2)

What I suggest to the authors is try to focus on one parameter at a time, explain the effect of treatment on that parameter then move on to the next paraments and if they wish to show the correlation between the parameters they can write this information in Discussion section or if they want to mention in result section they can do only after the complete explanation of the effect of treatments on all the given parameters.

Ø  Similar discrepancy was observed in results of ABA and ABA-GE also.

Ø  Another suggestion is since the authors are explaining all the stomatal related parameters in one paragraph and their correlations also (like stomatal conductance, Abaxial pore area, Abaxial stomatal width/length ratio, Abaxial Stomatal size, abaxial Stomatal density) so its better to merge Fig 1 and Fig 2 make just one figure and put all the graphs together and leaf temperature graph at the end. I feel it will be easy for the readers to see the effect of treatments on above mentioned parameters as well as the correlations among them that authors are trying to explain.

Rest of the manuscript is very well written, and focused and the hypothesis put on trial is also very interesting. So, what I suggest is, if authors incorporate the mentioned suggestions, It will further improve the quality of their manuscript.

Author Response

Dear Editor and reviewers:

We hereby send you the revised manuscript with manuscript ID: plants-2234500. Please note, that the manuscript was previously entitled “Effects of melatonin on stomatal regulation in tomato seedlings subjected to combined heat and drought stress”. However, the title have now been updated to “Exogenous melatonin alters stomatal regulation in tomato seedlings subjected to combined heat and drought stress through mechanisms distinct from ABA signaling”.

We appreciate the thorough review of our manuscript and the very constructive comments and suggestions. We have done our best to respond and consider the reviewers comments. Below we addressed all comments and suggestion from the reviewers. Our responses are in italic.

Respond to reviewer #2

➢ I would like to ask authors that have they did any seed viability test? I do not found any
information related to seed homogeneity and authenticity in material method section. If
they did kindly mention it in material method section.

We did not do any viability test. The reason being, that the seeds used were freshly brought for the experiment from the supplier, who ensured genetic homogeneity, high seed vigor and seed viability > 98%. Prior to our main experiment, we ran a few short trials with the same seeds to see the growth behavior of the cultivar, and the germination and of these 10 trial plants, did not give us any reason to doubt the statements of the supplier. But to ensure high homogeneity of the plants in the main experiment, we did actually sow around 60% more plants than needed for our experiment to be sure to have enough to discard any plants with differences in appearance or vigor. We added this detail to the method description 4.1 (L 406-408).

New sentence reads as follows:

At 11 days after sowing (DAS), after unfolding of the cotyledons and before formation of the first true leaf, 48 uniform plants were selected out of a total of 80 sown plants and moved to two separate PSI Fytoscope FS-WI walk-in climate chambers (Photon Systems Instruments, Drasov, Czech Republic). In the chambers, plants distributed on four 12-scale DroughtSpotter lysimeter units (Phenospex, Heerlen, The Netherlands)”

➢ Proline content is one of the most important stress markers but unfortunately it is not done by the authors, So I strongly recommend authors to include this parameter in their next study if they planned to work on stress.

Thanks for the advice. We agree that it would have been a good parameter to have included as a stress indicator in our experiment. We will look into our options for including that in future work.      

➢ As far as Result section is concerned the way authors explained the results are quite complicated to understand reason being authors were trying to explain and correlate so many parameters in single paragraph that may cause a difficulty for the reader to focus on
one particular parameter For example, Point 2.3 The results of Stomatal development and regulation. Under this heading authors not only mention the finding of stomatal conductance and leaf temperature but also mentioned the findings of stomatal width, pore area, stomatal width/length ratio and mentioned the correlations among all these parameters. (that are given in two Figures i.e Fig 1 and Fig 2).
What I suggest to the authors is try to focus on one parameter at a time, explain the effect of treatment on that parameter then move on to the next paraments and if they wish to show the correlation between the parameters they can write this information in Discussion section or if they want to mention in result section they can do only after the complete explanation of the effect of treatments on all the given parameters.

We highly appreciate this feedback on the readability of results section 2.3. Our reason for grouping data of stomatal conductance, stomatal anatomy and leaf temperature under one sub-section in the results is that there is a strong interplay between these data in a controlled setup like ours. We do acknowledge that the connection between the parameters could be made more clear to the reader to promote the understanding of our data. We therefore did following changes to the MS to improve this:

  • We added a few sentences to the introduction section to highlight the connection between stomatal anatomy, stomatal conductance and leaf temperature (L 63-69). This part now reads:

Stomata are pores in the plant epidermis, bordered by two guard cells with the ability to control the pore opening to regulate the gas exchange of CO2 and water vapor across the leaf surface. Stomatal properties such as, stomata size, pore area and stomatal density affects the stomatal conductance (gs) of the leaf [13]. When gs is high, transpiration can occur at higher rates to decrease leaf temperature through evaporative cooling [14]. One of the aspects of leaf stress physiology […]”

  • We added an explanatory sentence in the beginning of section 2.3 to set the theme of the section (L 180-182). This reads:

To evaluate the effects of melatonin and stress treatments on stomatal regulation, we measured gs and leaf temperature along with anatomical parameters of stomata at moderate and severe stress level.”

  • We restructured the existing text of section 2.3 to better group it into defined subthemes.

➢ Similar discrepancy was observed in results of ABA and ABA-GE also.

As for the comment above, we appreciate the feedback. To improve the readability of section 2.4, we have added an explanatory sentence in the beginning of section 2.4 and made some changes in the existing text (L235-237), and did some revisions on the phrasing of the text in section 2.4. The added sentence in the beginning of the text now reads:

“To investigate the hormonal regulation of stomata, the leaf contents of ABA and its conjugated form were quantified at moderate and severe stress level. Across treatments, ABA and ABA-GE were detected […]”

➢ Another suggestion is since the authors are explaining all the stomatal related parameters in one paragraph and their correlations also (like stomatal conductance, Abaxial pore area, Abaxial stomatal width/length ratio, Abaxial Stomatal size, abaxial Stomatal density) so its better to merge Fig 1 and Fig 2 make just one figure and put all the graphs together and leaf temperature graph at the end. I feel it will be easy for the readers to see the effect of treatments on above mentioned parameters as well as the correlations among them that authors are trying to explain.

It is a good point, and we do acknowledge that the layout of figures might require the reader to look between two figures when reading section 2.3. However, based on feedback on Fig. 2 from reviewer #1, we have updated Fig. 2 to include pictures of representative stomata of each treatment in a separate section. Adding data for stomatal conductance to Fig. 2 as well, would make it a figure with more too many elements for displaying at a reasonably scale for all elements. With this reasoning, we would suggest to keep Fig. 1 as it is and update Fig. 2 according to the suggestions of reviewer #1.

We believe that this prioritization of data representation is best in light of the improvements of the section 2.3 already described above.

We deeply appreciate your consideration of our manuscript, and we look forward to hearing good news from you.

Best wishes

Dr. Rong Zhou, Associate Professor from NAU

Email: [email protected] / [email protected]

Aarhus University

Agro Food Park 48, Aarhus N, Denmark
